# Exploring the Impact of Stigma on Health and Wellbeing: Insights from Mothers with Lived Experience Accessing Recovery Services

**DOI:** 10.3390/ijerph21091189

**Published:** 2024-09-06

**Authors:** Lydia Shrimpton, Michelle Addison, John Cavener, Steph Scott, William McGovern

**Affiliations:** 1Department of Social Work, Education and Community Wellbeing, Faculty of Health and Life Sciences, Northumbria University, Newcastle upon Tyne NE7 7XA, UK; john.m.cavener@northumbria.ac.uk; 2Department of Sociology, Faculty of Social Sciences and Health, Durham University, Durham DH1 3HN, UK; michelle.addison@durham.ac.uk; 3Population Health Sciences Institute, Faculty of Medical Sciences, Newcastle University, Newcastle upon Tyne NE2 4AX, UK; steph.scott@newcastle.ac.uk

**Keywords:** recovery, stigma, mothers who use drugs (MWUD), treatment, mental health, health services

## Abstract

Stigmatisation is the process by which an individual is devalued based on their attributes, characteristics, and/or behaviour, with this often leading to prejudice, social and health-related harms, active discrimination, and microaggressions. The aim of this paper is to show how social harms can occur and how stigma is damaging to the health and wellbeing of a person in recovery. To do so, we focus on the harms that arise from the internalisation of labels that mothers who use drugs encounter in a treatment and recovery setting whilst in active recovery, and how this stigmatisation can manifest negative self-beliefs. Qualitative data was used from two semi-structured focus groups involving females with lived experience of substance use (*n* = 13). A reflexive thematic analysis approach was used to analyse the interview transcripts, and three themes were identified: (1) the enduring nature of stigma; (2) gender disparity and the need for mothers- and women-only spaces; and (3) stigma as a barrier to services and wellbeing. Findings revealed the enduring nature of stigma amongst mothers who were in active recovery, with women feeling judged more harshly than men and experiencing pressure to live up to a “good mother” ideal whilst in recovery. This paper demonstrates that mothers in recovery are still stigmatised and, as a consequence, approach services with increased sensitivity, with stigma often resulting in disengagement or reluctance to access healthcare settings. We conclude that staff in health, social, and primary care settings need to develop a strong therapeutic alliance with mothers in recovery and promote anti-stigma approaches in their practice, in order to mitigate stigma and reduce harms to health and wellbeing.

## 1. Introduction

Stigma is a process by which an individual is devalued based on their attributes, characteristics, and/or behaviour [1]. Whilst marginalised groups can attract and experience stigma more intensely across societal, political, and relational situations, our specific focus in this paper is on people who use drugs (PWUD), and how stigma is experienced in drug recovery settings and internalised [2,3,4]. Although taking drugs is stigmatised, not all PWUD experience the same form or level of stigma [4], with different outputs initiating different stigmatising responses in society, such as what is being taken, e.g., injecting heroin [5,6,7,8,9,10,11], and who is taking them, e.g., mothers who use drugs (MWUD) [12,13]. It is also theorised that stigmatised attributes can intersect, with PWUD often also being stigmatised for mental health challenges, receiving benefits, and/or being open to social services [13]. MWUD are not only being stigmatised for their drug use but also for their mothering (e.g., whether they are a “good” or “bad” mother) [13,14].

Stigma is experienced in many ways and contexts, such as on an interpersonal level referring to stigma experienced from the public and in social settings [15]. Research has suggested that women who use drugs (WWUD), including those who are not mothers, are still subjected to interpersonal stigma regardless of where they are on their recovery journey, with women similarly stigmatised when using drugs and during recovery [16]. Family members can also be subjected to interpersonal stigma, with findings showing that families of PWUD are often attributed blame for the onset of their family member’s use, with this resulting in social exclusion [17].

Stigma is also experienced on an individual level where the stigmatised individual can experience self-stigma and internalise the labels they have been given, utilising them to create negative internal self-beliefs [18,19,20,21,22]. This can lead to feelings of shame and self-loathing [23,24,25,26]. Furthermore, this manifestation of stigma can lead to the stigmatised individual being less likely to engage with services and experiencing a reluctance to access treatment [21,27]. This reluctance to access treatment has also been associated with anticipated stigma, whereby the individual perceives that, when in service or seeking help, they will be judged and stigmatised by those from whom they are seeking help, therefore creating greater sensitivity when approaching support services and healthcare settings [28,29]. The impact of this can result in social harms, such as poor mental health and isolation from family and others around them [30,31].

Stigma can also occur on a structural level referring to institutional settings, such as within healthcare and provider-based contexts [15]. Statistics from 2022–2023 revealed that slightly less than one-third (32%) of people accessing drug and alcohol treatment in England were women. Of this, 27% of women in treatment were mothers [32], with this showing a clear gender disparity between men and women in treatment. This raises concerns regarding the number of mothers who are not accessing treatment, and the effect this may have on their own and their child’s wellbeing [33]. Similarly, findings from a study exploring stigma for women regarding drug use in a maternity setting showed that health staff judged and blamed WWUD when they failed to turn up for appointments or failed to adhere to treatment [34]. Previous findings have also demonstrated that the label of “former drug user” can be stigmatising for WWUD even when it is clear they are no longer using substances [16]. It is also widely known and accepted that stigma can create a barrier to treatment and recovery by creating either a pathway of secrecy for the person in recovery, followed by resilience and resignation in which they accept judgement against them, and/or fear in which they become anxious about accessing support [28].

Not only does stigma play an active role in inhibiting recovery and help-seeking behaviours [27], but it is also reported that experiencing stigma can result in women feeling less trusting of professionals [35]. This is seen in cases of MWUD, where a barrier to accessing treatment can be the fear and stigma of custody loss that is associated with accessing services [14], with child removal accompanied with marginalisation, trauma, and reduced social support [36,37]. Whilst there is a lot that is known about stigma as a barrier to service uptake and the social harms associated with stigma for MWUD, less is known about how MWUD and WWUD experience stigma during recovery, the implications of stigma in relation to social harms, and what actually can be done to reduce stigma and improve practice treating this group. The following paper demonstrates that mothers in recovery are still stigmatised by society and those around them, which can result in their disengagement or reluctance to access healthcare settings. We conclude this article by arguing that staff in health, social, and primary care settings need to develop a strong therapeutic alliance with mothers in recovery and promote anti-stigma approaches in their practice, in order to mitigate stigma and reduce the associated harms to health and wellbeing.

### 1.1. Study Design

A qualitative methodology was utilised to explore recovery experiences of MWUD. Two focus groups were conducted with people who identified as being in recovery and were accessing drug and alcohol and/or mental health services.

### 1.2. Participants

This study was conducted in the northeast of England as part of a pre-existing research project investigating criminal exploitation and serious violence during the COVID-19 pandemic. As part of this research project, focus groups were used to explore PWUD’s and MWUD’s experience of accessing services during the pandemic. Participants were recruited for the first focus group via the research teams’ pre-existing networks with a public-health forum involving service users across the city. An email was sent to forum members, informing them that a focus group, following their consent, would be conducted at their next forum session to explore exploitation and violence during the COVID-19 pandemic. A diversity of characteristics, past and present drug use, and consumption patterns were sought for inclusion in the study. Individuals were invited to participate in the study if they had some experience of drug recovery services and felt well and able to talk about their experiences of COVID-19 lockdowns between 2020 and 2022. Individuals were excluded from the study if they were below 18 years old. Snowball sampling was supplemented with purposive sampling in an additional focus group dedicated to exploring stigma in further depth for MWUD. The research team approached the lead of a peer-support group for those in recovery who have experienced child removal, requesting an expression of interest from those who wished to take part.

Thirteen (*n* = 13) females took part in two semi-structured focus groups (9f in focus group 1; 4f in focus group 2). Four (*n* = 4) males were also participants in focus group 1, with the data obtained from females in focus group 1 regarding motherhood and recovery isolated from the parent study for this research. Participants were comprised of women accessing services at varying stages of recovery, with some participants receiving Opiate Substitute Treatment (OST) whilst others were accessing services and groups to maintain long-term abstinence from all substances. All the women participating identified as mothers. At the time of interview, participants were defined as in recovery due to their accessing of drug and alcohol treatment services and recovery groups, and/or various non-affiliated services associated with recovery, e.g., female-only community hubs offering mental health and peer support to women in recovery. Snowball sampling [38] was initially utilised via the research team’s pre-existing networks, where participants were asked if they would like to take part in a research study exploring their experience of exploitation, vulnerability, violence, and accessing services during the COVID-19 pandemic.

Upon agreeing to participate in a focus group, the date and time were arranged. A Participant Information Sheet (PIS) was given to each participant pre-interview and an opportunity to ask questions was provided. A consent form was provided for each participant to sign prior to commencing the focus group. A verbal debrief was also given to the participants post-interview.

### 1.3. Focus Groups

Focus groups were conducted on 9 February 2023 and 16 November 2023 by two of the authors (L.S. and W.M.), who are researchers with previous experience interviewing health and social care professionals and working with service users in practice. The first focus group lasted 1.5 h, with breaks taken as needed. The moderators introduced themselves and explained the purpose of the focus group. The focus group questions were supported by a flexible topic guide developed by the research team using the previous literature and interim findings from semi-structured interviews with practitioners delivering services during COVID-19. Questions initially focused on challenges encountered accessing services during COVID-19; however, participants began to lead the conversation regarding barriers they had encountered when accessing services independent of the impact of COVID-19. These conversations were encouraged and supported by the facilitators and were steered by the mothers participating in the focus group relating to their experiences of being in recovery. A second focus group was organised to explore these findings further. The second focus group was conducted by W.M. and lasted for 1 h, following a similar format to the first focus group. In this focus group, a second flexible topic guide was used focusing solely on questions regarding participants’ experiences of stigma; their perceptions of stigma on individuals; how stigma can be a barrier to service engagement; and what factors practitioners need to consider. This second topic guide was developed following the first focus group and was informed by the findings related to stigma that organically transpired during the first focus group.

The focus groups were digitally audio recorded and transcribed verbatim. Participants were anonymised post-transcription and their responses were cleansed of any identifiable data. Field notes were also used to help explain the transcriptions and support analysis.

### 1.4. Analysis

Qualitative data analysis was used and followed a reflexive thematic analysis approach [39], in which interview transcripts were closely examined by the research team to allow for differing and comparable interpretations of data. Due to the research being exploratory of MWUD’s experience of being in recovery whilst accessing services, the study design was informed by an interpretivist paradigm [40]. As interpretivism assumes that reality is subjective, multiple, and socially constructed, the researcher was able to analyse the data with an understanding that one participant’s experience may be different from another and is shaped by their perspective, ensuring that what is produced is reflective of the participants’ reality [41]. The researcher’s positionality when leading the first focus group was as a woman with previous experience working in safeguarding and drug and alcohol settings, which supported a reflexive and empathetic approach to fieldwork. The researcher’s positionality for the second focus group was as a male with previous experience working in clinical settings, and who was known to the participants due to previous public involvement and community engagement work. It was recognised that participants generally felt more comfortable sharing their experiences with a female researcher due to past experiences of violence and trauma; however, participants were still able to be open with the male researcher due to a strong rapport already having been established prior to the interviews. The focus group was carefully and sensitively facilitated, allowing a supportive and trusting group dynamic to develop. The researchers also disclosed their insider status regarding past professional experiences working with PWUD in drug recovery services. This had the effect of allowing language and acronyms referring to drug types, uses, and patterns to flow more quickly in the group, leading to shared understandings and a feeling of trust.

Analysis resulted in the identification of common themes and ideas across the transcripts. This analysis involved six stages: (1) data familiarization, (2) code generation, (3) theme generation, (4) theme review, (5) theme definition, and (6) result production. Each transcription was checked by this article’s first author (L.S.) against the digital audio recordings, prior to deletion, to ensure accuracy.

## 2. Results

The following key themes emerged from the qualitative data: (1) the enduring nature of stigma; (2) gender disparity and the need for mothers- and women-only spaces; and (3) stigma as a barrier to services and wellbeing. These themes are outlined below.

### 2.1. The Enduring Nature of Stigma

This theme describes the ways in which participants continued to experience stigma as they progressed into recovery. Mothers in the focus groups highlighted the ways in which they were stigmatised as they engaged with different services and organisations. 

There was a general feeling amongst participants that stigma was not reduced and was felt significantly more after they had stopped using drugs and began to move towards recovery:

*“I felt just as much shame about being in recovery, outwardly to society, as I did when I was in active addiction. When I was in active addiction, I didn’t feel it as much”*. (Female in recovery 4, Focus Group 2)

Perceptions about shame, and being shamed, resulted in those mothers who were seeking help feeling that they had to do so secretly and in isolation. During the focus groups, mothers described a range of specific situations in which they were actively taking steps to move towards recovery and improve their situation to look after their children. They perceived that, during this time, they were being stigmatised because of their situation, and that this was reflected in the ways in which professionals behaved towards them. In the quotation below, the participant describes how she felt judged and ashamed in a hospital setting following childbirth:

*“There’s stigma in the hospitals as well, d’you know, like the staff sniggering, like other parents […] hearing nurses like sort of speak down corridors […] like when you’re not coming in, or you’re given your methadone, or your kids are withdrawing on the special baby unit, and, having to answer to society, like why is your kids still in hospital […] Feeling like you just want the ground to swallow you up and just not having the answers and not wanting to post that you’ve gave birth, ‘cos you know that people are going to be like, “oh, why is she still in?”* (Female in recovery 4, Focus Group 2)

Mothers also explained how they found themselves in situations where they were stigmatised despite doing what they felt was best and correct for themselves and their children. Almost half of the mothers in this study had their child taken into care, and despite recognising that this was the correct decision for them and their children, they had still experienced stigma and shame for losing their children—being seen as a “bad mother”—and/or for not trying hard enough to maintain custody of their children:

*“People [say] “Why did you just give up on your children?” Well actually, I didn’t, you know? And that’s the stigma that I get. Like, “why did you just give them to their dad?” Like, you know? I lost my children… But then in my instance, I gave my children up because I knew I couldn’t look after them properly. I couldn’t provide the stability, the routine, the structure and… You know, they’d never… They were never short of love. You know? But it was just because I couldn’t adult and I couldn’t parent, because my drug use was off the scale, so I did the kindest thing for them, which meant give them to the father and me run away, so I feel a lot of stigma around that”*. (Female in recovery 2, Focus Group 2)

Mothers also highlighted how they were labelled as “trouble”, were judged, and had assumptions made about their personality and character because they were known to have previously used substances. Mothers in this study spoke about how other parents and professionals acted towards them in this manner:

*“So, they sent the report to the nursery, and I wasn’t drinking at the time, and they pulled us aside… Like, [describing situation] all the parents used to go into the waiting room and wait for their kids and pick them up, and the head teacher pulled us aside one morning after drop-off and said, “some of the parents have reported that they could smell alcohol this morning” and I was like “it’s not me”. […] “Yeah, [they replied] but we got a report from social services”, and I know she was, like, doing her job, but I mean really…”*. (Female in recovery 5, Focus Group 1)

These discussions reveal how those in recovery still experience and endure stigma, and in some cases feel the effects of stigma more intensely. This was reflected in how mothers, despite taking steps towards recovery for the best interest of their children, still experienced negative labelling and situations as a result of their history and current use. 

### 2.2. Gender Disparity and the Need for Mothers- and Women-Only Spaces

There was a shared agreement amongst focus group members that women and mothers in recovery were judged more harshly by professionals and others in recovery in comparison with men. Focus group members spoke about their perceptions of what it means to be a woman who currently uses/has used substances:

*“I think as a woman with addiction, it’s very hard when you’ve got children, because we’re judged a lot harsher than men, because we’re supposed to be women and mothers and… So, like I said, with me, I was terrified when I relapsed, to go to the doctors, because I knew social services would be brought back in. My son was taken off us after, but as a woman and a mother, I think we’re judged more harshly, and we need a lot more help”*. (Female in recovery 6, Focus Group 1)

The quotation above illustrates that participants perceive fathers to be able to access services with more ease, and with limited negative consequences in comparison with mothers. Mothers also described stigma as direct and perpetuated by practitioners via negative language, such as the term “neglect”, with negative and blaming language found to negatively affect a person’s willingness to access treatment and engage with services [28]:

*“You’re given the same words and the same language if you’re a mother who’s struggling with addiction, as a perpetrator that has purposely harmed their own children. It’s the same language, it’s the same way it’s described and it’s not the same. So that’s stigma there”*. (Female in recovery 4, Focus Group 2)

Mothers also explained that a lack of awareness around stigma and limited discretion from professionals meant that they were often exposed to labelling, negative experiences, and shame. This was also highlighted as perpetuated by other people in recovery. In the quote below, a mother described how she felt labelled and judged because she had “lost her kids”. She continued by discussing what she described as the toxic culture associated with hyper-masculinity that surrounds particular types of substance use and recovery groups [15]:

*“Outside of recovery but inside of recovery as well, because, like you said before, there aren’t a lot of females in 12 step fellowship, you know? And it is quite a masculine environment and it’s kind of like… It’s kind of a sort of acceptable for men to say, “oh, I lost my kids” but when a woman says, “I lost my children…” […] which meant give them to the father and me run away, so I feel a lot of stigma around that”*. (Female in recovery 2, Focus Group 2)

Overall, mothers in this study reported that they benefitted from seeing other mothers present in recovery services, and from attending gender-specific recovery settings. During the focus groups, participants identified that they felt inspired and encouraged by the presence of other mothers and women in active recovery. Mothers also spoke about feeling positive as they were able to identify with others, highlighting how women-specific recovery groups allowed for conversations around a shared understanding and experience of stigma:

*“If more women started… Did feel comfortable to open up in meetings, then other women would be able to know that they’re not alone. Because that’s what I felt like; I felt like I was the only person […] coming to these groups and meeting other women; it’s just… It’s inspirational”*. (Female in recovery 1, Focus Group 2)

Participants further identified feeling that women and mothers were required to work harder to be in recovery, as a result of gender-specific stigmatising processes, such as the “good” and “bad” mother ideal. They noted that stigma was a concern they faced in its own right, with it manifesting itself into negative self-myths, leading to mothers having to overcome stigma as a part of their own personal wellbeing goals.

In general, theme two presents evidence for how people in recovery perceive women/mothers to experience challenges when accessing and engaging in recovery, with there being the added expectation to uphold the “good mother” ideal whilst seeking support. It was highlighted that this extra pressure was not seen in fathers seeking treatment, with it suggested that it was easier for a man to access recovery.

### 2.3. Stigma as a Barrier to Services and Wellbeing

A small number of participants in this study reported that their previous experiences of stigma had hindered them from seeking professional support and assistance. It was suggested that previous experiences of stigma can lead to those in recovery feeling unable to seek support at a consequence of poor mental health and low self-esteem:

*“It teaches you to keep your mouth shut and I think that’s where a lot of mental health kicks in”*. (Female in recovery 1, Focus Group 2)

Previous experiences of stigma and shame also led mothers to perceive that they would be stigmatised further by others and health care professionals, with this resulting in some mothers avoiding certain social situations and not seeking out activities to do with their children. The participant below described the stigma associated with using drugs and having a child, with this leading them to feel unable to take part in mother and baby activities for fear of being judged:

*“And not being able to go into like, baby groups ‘cos you’re having to give oramorph, you know, and people asking questions. I think as a parent, if you’re lucky enough to take your kid home, then it’s a no-win situation; it you get them took as well, in the hospital, you know, that stigma and shame and if you get to take them home, you know, that stigma and shame, it’s just riddled with it”*. (Female in recovery 4, Focus Group 2)

This can escalate into fear of approaching services due to concerns of judgement and the potential negative repercussions for seeking help:

*“Well, if I’d went to social services and said, ‘look, can you get us help with me mental health’, they would have took [child’s name] before I’d even relapsed. So, it’s like you’re scared to get any help. Even doctors… Anyone that you mention to that you’ve got children like, ‘I need help’, social services will get involved straight away, so you’re stuck, really. Like, who…? You’re scared to talk to anybody”*. (Female in recovery 5, Focus Group 1)

It is not unusual for mothers in substance use contexts not to trust services. There was a general agreement in this study amongst participants that they perceive some services to work against them as opposed to with and for them. This can lead to those in recovery feeling they have to present in a certain way to avoid judgement and guard themselves when seeking support for health-related concerns. This can also lead to harms as mothers fail to disclose their needs and concerns due to fear of negative repercussion and previous stigmatising experiences being replayed. Mothers described how they found it difficult to engage with their workers regarding their own physical and emotional wellbeing and how they had previously downplayed or hidden their own needs:

*“I went to the doctor’s when he was a month old, ‘cos he had really bad reflux and I have carpal tunnel in me hands now, ‘cos he’s so heavy, so I’ve got bad wrists and I’ve had to go to the doctor’s and was terrified. And I made sure I smelled nice, I done my hair… I put a bit of makeup on because I was absolutely petrified and then I was like, ‘don’t cry, don’t cry, don’t cry…’ which is ridiculous. But it’s just the way I feel because I don’t want to miss anything [because I live on my own, so obviously I’m that part of the childhood], you know what I mean, but it’s still in the back of my mind every time I go to the doctor’s”*. (Female in recovery 6, Focus Group 1)

The participant above also disclosed that during pregnancy, she hid the physical pain she was experiencing for fear that she would receive negative consequences for seeking help.

A concern raised by participants was that stigma in services can also lead to them being denied specific help and further labelled as too complex to be engaged in some services:

*“It’s a vicious cycle. A lot of the time you go for help and you’re using, and they say, ‘we can’t help you with your mental health until you kick the drugs side of it…’”*. (Female in recovery 7, Focus Group 1)

This theme highlights how stigma can create a barrier to those in recovery from seeking health-related support in an open and honest manner. It displays how mothers in recovery often feel they must guard certain health concerns to avoid negative consequences and labels.

## 3. Discussion

The findings in this paper provide deeper insight and reveal the enduring nature of stigma, with MWUD still experiencing stigma, both interpersonal and structural, regardless of the stage of recovery they are in. This stigma manifests itself at an individual level as self-stigma and shame, with this leading to anticipated stigma as the person in recovery becomes fearful of accessing services and feels isolated in their recovery. Rather than simply reinforcing the idea that MWUD experience social harm as a result of stigma, this paper moves on to discuss the social harms associated with stigma for mothers in recovery and those accessing services. Findings also show that the effects of stigma lead to those in recovery approaching services with a level of anxiety, resulting in their main support being found in their peers with lived experience. Findings also highlight that WWUD and MWUD are perceived to be judged more harshly than men, with more negative repercussions experienced when accessing services where they are labelled as harming their children, and stigmatising attitudes in healthcare settings being demonstrated towards them.

The findings that those in recovery still experience stigma, both interpersonally and structurally, are consistent with the previous literature reporting stigma to be enduring [42], with preconceived judgements and current and historic information regarding drug use found to inform how others treat those in recovery [28]. This endurance of stigma throughout recovery is also seen when PWUD relapse, with this being perceived as negative, leading to public and self-stigma [43]. Relapse during recovery has been noted as incremental to change. The Transtheoretical Model argues that habitual behaviour, such as addiction, is not changed quickly, but occurs through a cyclical process, with relapse an important stage to achieve long-term change [43,44]. As evident in this research and in previous findings, stigma perpetrated towards those in recovery can lead to social harms and manifest as negative self-judgement, becoming a barrier to accessing health-related services [21,27]. This is particularly evident where participants report feeling shame because of the stigma they have experienced and feeling fear when approaching services [2]. Anticipated stigma can be as significant as actual stigma, with people in recovery approaching services with the expectation that they will be stigmatised [28].

Our findings also indicate a gender disparity regarding stigma, with MWUD receiving harsher judgement in comparison with fathers generally and when accessing treatment. This is in line with double deviance theory, in which women involved in the criminal justice system as both victims and perpetrators are more likely than males to experience stigma for their violation of social norms and expectations as women [45], or in this study’s case, as MWUD. Findings from this paper are also consistent with the literature that finds MWUD experience the additional pressure of fitting into the “good mother” ideal [13,14]. However, they also suggest that mothers and women who use substances need their own “gendered spaces” where they can either discuss and articulate their own concerns with other mothers and women, or where they are able to fully articulate their concerns with practitioners who are both sensitive and vigilant to their previous experiences of stigma and current and ongoing needs [13].

A reluctance to seek support and access services can be detrimental and harmful to the health and wellbeing of the MWUD, and stigmatising responses from healthcare professionals can lead to disengagement from services and a further reluctance to re-enter into services when needed. This ultimately has a detrimental impact on the MWUD’s physical health, as they are less likely to access health services when required [21,27]. A secondary effect of this is the harm that not accessing drug and alcohol treatment can have on the children in the household experiencing familial substance use without intervention, with the children experiencing an increased risk of child maltreatment, social service involvement, and engagement in substance use themselves [46]. It can also have a negative effect on MWUD’s mental health, with stigma, even anticipated stigma, associated with low self-esteem, depression, and anxiety [47]. These findings are similar to those in which postpartum participants using drugs described and internalised negative comments and looks by hospital staff [28]. They found that this stigma impacted how they felt they were perceived by clinicians, as they felt that those caring for them judged them and believed them to be “drug-seeking” when they sought medical support. The fear that MWUD reported in this study of accessing health settings due to prior experiences of child removal, and their fear of it happening again, is indicative of how traumatic child-loss can be, with child removal associated with trauma, grief, depression, and stigma [37]. This further explains the fear participants had of this happening again if they were to access a service, creating a reluctance for MWUD when accessing healthcare settings where they have experienced stigma before. This is consistent with the literature exploring MWUD who were also involved in the criminal justice system. In these findings, it was revealed that MWUD experienced stigma, which became the narrative of their motherhood, with MWUD experiencing shame and feeling they had “failed” at being a mother. This stigma similarly resulted in a reluctance to access services due to limited trust with professionals [12].

### Strength and Limitations

Although the research sample was small in number, a strength of the research was the research team’s pre-existing professional relationships with the participants due to previous public involvement and community engagement sessions for other research projects. This enabled the participants to feel more comfortable when sharing their experiences about stigma in an open and authentic way, providing enriched data. However, it is worth noting that, due to the researchers’ existing relationships with participants, the researchers were already informed on the women’s history of drug and alcohol use and being in recovery, therefore introducing the potential for confirmative bias. That being said, precautions were taken during analysis to discuss and examine emerging themes arising from the data with the wider research team.

Due to the nature of the discussion of stigma in the first focus group occurring organically and separately from the initial project, there was no interview guide followed in focus group 1 for conversations relating to stigma; therefore, different questions were asked in each focus group, resulting in potential gaps in what was explored in each focus group compared with the other. Because the theme of stigma and mothering was an emergent finding, the original study did not collect demographic information relating to time in recovery, children, and how many children had been removed from their care, other than what was stated during the focus groups. As such, we were unable to compare factors such as time in recovery or child removal with experiences of stigma; we see this as a line of enquiry for future research. Inclusion criteria for this research project focused on PWUD. A diverse set of identity characteristics, consumption patterns, and drug use were sought, although we excluded anyone below 18 years old. The project was inclusive of all individuals who identified as a woman; however, we did not explore this as a theme within the study; these findings are therefore not necessarily reflective of experiences held by all marginalised individuals who identify as a woman.

## 4. Conclusions

From the findings, it is evident that stigma is a barrier for those in recovery when accessing health and social care services and has detrimental impacts on health and wellbeing. Stigma creates apprehension when approaching services due to previous experiences of discrimination in similar settings, leading to limited trust in relationships with practitioners, and a guarded presentation from those seeking help. The stigma experienced in healthcare settings highlights a need for practitioners to develop a strong therapeutic alliance with people in recovery, and to promote anti-stigma approaches in their practice to mitigate stigma from colleagues. As part of this therapeutic alliance, it is recommended that practice be trauma-informed, and thereby practitioners adopt principles of ensuring safety, developing trust, and working collaboratively with the PWUD, as well as applying anti-stigma interventions and understanding how the role of “mother” has become a stigmatised identity for MWUD. Findings from this study also stress a requirement for practitioners to analyse their own stigmatising behaviour—whether this be intentional or unintentional—with a view to delivering non-stigmatising services. Incorporating these anti-stigma interventions will help MWUDs to feel more comfortable accessing health-care settings, diminishing the fear of being stigmatised and being encouraged that practitioners are striving to achieve an anti-stigma approach. From a public health perspective, such interventions have the potential to lead to more women accessing recovery services and support, with findings highlighting anticipated stigma from previous experiences on an individual–practitioner level and a service-based level to be significant barriers to accessing current recovery settings. We also suggest that services consider creating women- and mothers-only spaces and recovery groups, with findings demonstrating the benefits of peer support and visible recovery amongst mothers, and showing particular challenges, such as gender-specific stigma regarding child removal, experienced by MWUD and WWUD. We recommend that future research on stigma experienced by MWUD is collaborative in its approach, and that planning and decision making incorporate key players in the community, such as policy makers, commissioners, service providers, practitioners, and those with lived experience. This will encourage a wider reflection and understanding regarding the mechanisms of stigma and the impact stigma has on MWUD [15].

## Data Availability

Due to the nature of this research, supporting data is not available..

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
