# Peer review of "Exploring the Impact of Stigma on Health and Wellbeing: Insights from Mothers with Lived Experience Accessing Recovery Services"

_ijerph, 2024, doi:10.3390/ijerph21091189_

Round 1

Reviewer 1 Report

Comments and Suggestions for Authors

Thank you for the opportunity to review this manuscript that is a qualitative investigation of the impact of stigma and the health and wellbeing of mothers who have accessed recovery services. The study findings offer important information on, and insights into, women’s experiences with intersectional gender- and drug use-related stigma and how that relates to their use of, and experiences with, recovery services. There were several strengths including the richness of the qualitative data, however I do have a few suggestions and concerns:

o   It is recommended that the authors address and specify what is meant by “women” and the issues surrounding sex/gender in this paper. Are transgender women/mothers or gender diverse persons included in this study? Or were cisgender women, specifically, the focus of this analysis? If the research was restricted to cisgender women, it is recommended that the authors provide a rationale/justification/explanation for this and an acknowledgement of this within the limitations section. Overall, the manuscript would be strengthened if the authors spent time addressing the sex/gender issue and what this means for our understanding of women’s and mother’s experiences with stigma and recovery services.

o   Overall, the Introduction and Discussion sections could use some streamlining and a thorough readthrough. For example, on page 2, line 28, the authors state “Statistics from 2022-2023 revealed that more than two-thirds (32%)…” – but 32% is a little less than one-third. Also, though the authors do define stigma and discuss stigma processes, the paper would benefit from the manifestations of stigma (e.g., anticipated, internalized, enacted/experienced stigma) being more well-organized and clearly distinguished/defined as well as the different levels in which stigma can occur (e.g., structural, interpersonal, and individual level) – and what is known about these manifestations and levels of stigma as they relate to women who use drugs. In addition, do the authors expect that the experiences of stigma may differ for women in recovery compared to women who use drugs in other settings? Why or why not? This could help better highlight the gaps in the existing literature and better underscore what the study findings contribute to this work.

§  For example, on pg. 2 lines 6-9, the authors delineate that the focus of the current paper is PWUD and how stigma is experienced (and internalized) in drug recovery settings – but then lead into a broader discussion of prejudice and micro-aggressions and internalized stigma. It is recommended that the authors, instead, focus on what is known about internalized stigma for PWUD, specifically, here.

§  There is also a need for consistency in the terms used across sections (e.g., in the introduction the term anticipated stigma is used, and in the discussion the term perceived stigma is seemingly used to indicate anticipated stigma).

§  Also, the study findings, to me, seem to highlight how structural-level gender- and drug use-stigma (e.g., in policies and institutions) is experienced by women. Given the discussions related to child removal/giving up children, social service involvement, hospitals/health care institutions, the lack of women-specific spaces, etc. – might it also be useful to better highlight and emphasize the role of structural-level stigma (in addition to individual and interpersonal stigma) in the lives of women accessing recovery services throughout the paper? Since I’m only privy to isolated fragments of participant narratives – I defer to the study authors on whether this feels like an accurate representation of the qualitative data collected. It just appears to me that structural forces play a prominent role in women’s experiences of stigma here.

·        Since much of this paper highlights the intersection of gender & drug use – and given that there are several structural-level forces that seem to be shaping women’s experiences with stigma – intersectionality theory/intersectional stigma might be a useful way to help frame this work. It seems like the authors allude to the concepts within intersectionality within the introduction but don’t use the corresponding term(s) and don’t cite the relevant intersectional literature – so I will also defer to study authors on whether this framing feels appropriate for their work.

o   Greater detail is needed in the methods section regarding study procedures and participant recruitment. In particular, it would strengthen the paper if additional information was included regarding the inclusion and exclusion criteria of both the parent study and this analysis.

§  The authors stated that, at the time of interview, participants were accessing drug and alcohol and/or mental health services. For those who were solely accessing other mental health services – how did you determine/operationalize being in recovery/recovery service use?

§  In addition, of the 13 women that were the focus of the current analysis – how many were mothers? The framing of the paper makes it seem like all women were mothers – was this true? It would be helpful to specify this information in the methods section.

§  More information on the rationale for, and linking of, the two separate topic interview guides is needed.

o   Throughout the paper, the authors describe anticipated stigma leading to “hypersensitivity” when seeking support services for women who use drugs. Is there a better term for this? Do we have any evidence that women are being overly sensitive within these services? Or are they having a normal response to discriminatory systems and people?

o   I also wonder if there is a better way to phrase the following statement (pg. 8, lines 8-9): “… to discuss the social harms associated with stigma for people in recovery and those who are actually trying to better their situation.” The current phrasing sounds potentially judgmental of PWUD who are not currently in recovery.

Comments on the Quality of English Language

The paper was well written and easy to follow, but would benefit from minor editing to improve the readability.

Reviewer 2 Report

Comments and Suggestions for Authors

Thank you for the opportunity to review this paper. I thought it was really interesting and adds to the evidence base for the need for women only spaces in addiction services and the need for anti-stigma work for women and especially MWUD.

1.1 Study design and 1.2 Participants

Report that focus groups were conducted with PWUD. Can you clarify if that means that men were included in the groups and you have only used the data from women or did the focus groups only contain women. Similarly, did the women include only MWUD or not.

1.4 Analysis

Missing information about epistemology or theoretical position used. Also would be useful to have a section on researcher characteristics or positionality to understand the research teams position, potential areas for bias and how this was considered in the analysis. It would be interesting to hear your reflections on interviewing women and mothers about stigma – given your roles and the power dynamics plus any possible impact of gender of interviewer.

2. Results

It would have been useful to have some information on the participants – such as age, as the point above were they all mothers, number of children, having children in care. Your quotes describe all the women as female in recovery with a number and which focus group. But in the participants section you stated they were in various stages of recovery. Is there more information about what these stages are?

Almost half of the quotes came from one participant – 4, focus group 2. Is it possible to include quotes from the other women either in addition or to create some balance?

3. Discussion

It would be useful to have a strengths and limitations section in this section. You briefly mentioned that women avoid accessing treatment but it would be good if you could expand on this about the risks to women and their children from this avoidance and the impact this might then have on women but also on social work and child protection/welfare issues.

4. Conclusion

I wonder if you’ve undersold your results a bit here and in addition to the anti-stigma work needed at an individual practitioner and service based levels, there’s also scope for wider anti-stigma work as a public health issue which could have an impact on all three of your themes.
